# Fish Liver Discards as a Source of Long-Chain Omega-3 Polyunsaturated Fatty Acids

**DOI:** 10.3390/foods11070905

**Published:** 2022-03-22

**Authors:** Charlotte Jacobsen, Simone Andrea Warncke, Sussie Hjorth Hansen, Ann-Dorit Moltke Sørensen

**Affiliations:** National Food Institute, Technical University of Denmark, DK-2800 Kongens Lyngby, Denmark; simonewarncke@gmail.com (S.A.W.); sussiehjorth@hotmail.com (S.H.H.); adms@food.dtu.dk (A.-D.M.S.)

**Keywords:** hake, ling, cod, monkfish, coalfish (saithe), omega-3 PUFA

## Abstract

The intake of omega-3 polyunsaturated fatty acids (PUFA) of the average consumer is generally low, and products such as fish oils high in omega-3 PUFA have become popular dietary supplements. There is a need for more sources of omega-3 PUFA to cover the increasing demand. This study investigated whether livers from different lean fish species could be a potential new source of oils rich in omega-3 PUFA. The seasonal variation in lipid content, fatty acid composition, peroxide value and free fatty acid content (FFA) of livers from cod, hake, ling, coalfish and monkfish was determined, and the effect of storage conditions on the fishing vessel (ice vs frozen) was studied. Generally, the lipid content and composition of the livers from the five fish species varied similarly during the two years of the sampling period, with significantly lower values in spring (March, April) and higher values in fall (October, November). Storage conditions were found to have no significant effect on the quality and oil composition. Monkfish livers were less suitable for production of omega-3 oil due to their lower lipid and EPA content as well as higher FFA levels. Coalfish had higher fluctuations in oil composition during the sampling period, which potentially makes a standardised quality difficult to obtain. Cod, hake and ling were the most suitable species for fish liver oil production.

## 1. Introduction

It is commonly known that most products of marine origin are essential in a healthy diet because they are rich in long-chain omega-3 polyunsaturated fatty acids (PUFA). The important omega-3 PUFA in the diet are eicosapentaenoic acid (EPA; C20:5 n-3) and docosahexaenoic acid (DHA; C22:6 n-3). Consumption of omega-3 PUFA contributes to maintenance of health and significantly reduces the risk of cardiac related diseases and stroke [1,2]. Omega-3 PUFA are also important for the neurological functions, as EPA and DHA are components of brain and nervous tissue and, therefore, are important for neurological development in infants and also for maintaining mental health [2,3].

Omega-3 oil supplements, especially in encapsulated form, are gaining increasing popularity, as a way for the average consumer to reach an intake that lives up to the daily requirements of EPA and DHA as recommended by different organizations such as EFSA. Most of the omega-3 oils used for dietary supplements are imported from the two largest exporters, Peru and Chile, which account for, respectively, 31% and 15% of the total world production of fish oil [4]. These fish oils are mainly produced from sardines and anchovies, which are also used to produce fish oil for fish feed. The production of farmed fish and, thereby, the need for long-chain omega-3 PUFA for both fish feed and for human consumption, is increasing. However, it is not possible to increase the catch of anchovies, sardines and other fish species used for fish oil production due to sustainability issues. Therefore, other sources of long-chain omega-3 PUFA are urgently needed.

Up to 44 million tons worldwide of unutilized fish raw material is discharged annually, meaning that there is a huge potential in taking advantage of this waste and, for example, use it in the production of marine oils [5]. Previous studies on the potential of utilising fish waste have mainly focused on fatty fish (>8 g fat/100 g) with known high PUFA levels, such as salmon, trout and anchovies [6,7,8].

White fish live on or near the seafloor, in contrast to the oily or pelagic fish, which live away from the seafloor. White fish have flakier white- or light-coloured flesh. White fish species are lean (<2 g fat/100 g) and low in fat in the flesh, but deposit a high content of oil and fat in the liver. For example, lipid content varied from ca. 40% to 60% of wet weight in Baltic cod [9]. Cod liver is already used for extraction of omega-3 PUFA in some parts of the world [4]. Ling, coalfish (saithe), hake and monkfish are other examples of white fish species, which are caught for human consumption in Scandinavia. On board the fishing vessel, viscera including livers are removed from the fish and discarded back into the sea because of the low economic value of these parts of the fish. Moreover, viscera and liver are rich in enzymes, which potentially can degrade the fish muscle and, thereby, reduce its quality. According to Statistic Denmark, approximately 33,000 and 22,000 tons of cod species were caught in Denmark in 2018 and 2020, respectively [10], and 40% of the fish are utilized for consumption after trimming. This number could be increased by 10% by utilizing the liver, and even more by including all viscera.

Due to the unsaturated nature of omega-3 PUFA, they are highly susceptible to lipid oxidation. This could be a problem if livers are to be brought back to the shore for further processing. Likewise, lipases present in livers may degrade triglycerides and phospholipids and produce high levels of free fatty acids.

The aim of this study was to evaluate the potential of livers from five different species of white fish (ling, coalfish, hake, cod and monkfish) caught in Danish waters of the North Sea for use as a new source of omega-3 oil. This includes investigation of the seasonal variation in the lipid content and fatty acid composition of the livers with the focus specifically on the content of EPA plus DHA and total content of omega-3 and omega-6 PUFA. Furthermore, the aim was to evaluate whether the storage temperature on board the fishing vessel (ice storage versus frozen storage) will affect the oxidative status measured by the peroxide value (PV) and lipid hydrolysis as measured by the content of free fatty acids (FFA).

## 2. Materials and Methods

### 2.1. Raw Material

For this study, fish livers from five fish species (ling (*Molva molva*), monkfish (*Lophius piscatorius*, coalfish also called saithe (*Pollachius virens)*, hake (*Merluccius merluccius)* and cod (*Gadus morhua*)) were obtained from two vessels, HG 306 Tobis (Hirtshals, Denmark) and HM 635 Karbak (Hanstholm, Denmark), throughout a period from August 2016 to July 2018 (samples from Tobis only received in August 2016). An overview of the samples obtained is shown in Table 1. The fish were caught in the North Sea, mostly along the southern part of the Norwegian coast. The livers were removed from the fish before being stored on the vessel either on ice or frozen at −20 °C for up to six days. When received at the harbor of Hanstholm (Denmark), they were picked up by a truck and transported at −20 °C to the Technical University of Denmark. Upon arrival, the livers were transferred to a freezing room at −40 °C where they were kept until analyses were performed. For each species, up to seven livers were sampled per fishing trip. In most cases, two to four livers were collected, and results are reported as means ± standard deviation. For each liver sample, all analyses were performed in duplicate.

### 2.2. Methods

#### 2.2.1. Dry Matter

Dry matter of the fish liver was determined after chopping according to the AOAC Official methods of analysis [11].

#### 2.2.2. Oil Content

Oil content in the fish liver was determined in chopped solid samples according to Bligh and Dyer (B&D) [12], using a reduced amount of solvent but keeping the ratio between water, methanol and chloroform. Results are presented as % of the total sample.

#### 2.2.3. Fatty Acid Methyl Ester (FAME)

The FAME profile of the fish liver was determined by analysing samples of B&D extract corresponding to 20–50 mg lipid. Fatty acid profile was determined based on the official methods Ce 1b-89 [13] and Ce 1i-07 [14] of the American Oil Chemist’s Society (AOCS), with some modifications. Approximately 2 g of B&D extract was weighted in methylation glass tubes and evaporated under a stream of nitrogen until dryness. A mixture containing 100 μL of internal standard solution (C23:0), 200 μL of heptane with BHT and 100 μL of toluene was added to the dry extract. Samples were methylated in a microwave oven (Microwave 3000 SOLV, Anton Paar, Ashland, VA, USA) for 5 min at 100 °C and power of 500 W. After methylation, heptane with BHT (0.7 mL) and saturated salt water (1 mL) were added. The upper phase (heptane) was transferred into HPLC vials and analysed using gas chromatography (HP5890 A, Agilent Technologies, Santa Clara, CA, USA). Fatty acid methyl esters were separated by the GC column Agilent DB wax 127–7012 (10 m × 100 μm × 0.1 μm) (Agilent technologies, Santa Clara, CA, USA). A standard mix of fatty acids methyl esters (Nu-Check-prep GLC 714, Nu-Check Prep. Inc., Elysian, MN, USA) was used for fatty acid identification. Fatty acids were quantified as area % of total fatty acids.

#### 2.2.4. Free Fatty Acids (FFA)

The FFA content was determined according to AOCS Official Methods Ca 5a-40 [15] by titration, with 0.1 M NaOH on samples consisting of 10–15 g B&D extract mixed with 20 mL chloroform, 25 mL ethanol and 5 drops of phenolphthalein indicator.

#### 2.2.5. Peroxide Value

Peroxide value was determined on the B&D extract using the method of Shantha and Decker [16]. In brief, 50 µL ammoniumthiocyanate-solution (30%) and 50 µL iron (II) chloride-solution were added to an amount of B&D extract, corresponding to 0.02 g–0.1 g of lipid. The iron (II) chloride solution was prepared from 0.25 g FeSO_4_ * 7 H_2_O in 25 mL H_2_O and 0.20 g BaCl_2_*2 H_2_O in 25 mL H_2_O. The peroxides oxidized the ferrous ions to ferric ions, which then reacted with thiocyanate to create a red-coloured ferric-thiocyanate complex, which was determined spectrophotometrically at 500 nm. Results are presented as meq. peroxides (ROOH)/kg oil using iron (III) for the preparation of standard curves. Stock solutions of Iron (III) was prepared as follows. Approx 0.5 g FeCl_3_ * 6H_2_O was dissolved in 50 mL 10 M HCl and 1–2 mL H_2_O_2_ was added. Then, the solution was boiled for 5 min and cooled to room temperature, whereafter it was diluted to 500 mL with H_2_O.

#### 2.2.6. Data Analysis

Results are reported as average and standard deviation. Multiple sample statistics was performed using Statgraphic (Version 18.1.06, Statpoint Technologies, Inc., Warrenton, VA, USA), followed by Tukey’s post-test to identify significant differences between the sampling points within a species. Significant differences are denoted with different letters in superscript within the same column of a table. A significant level α = 0.05 was applied.

To get an overview of the differences between months and species for all analysed parameters (dry matter, oil content, FAME profile, free fatty acid content and peroxide value), a principal component analysis (PCA) was conducted using the Unscrambler^®^ X software (Camo Software AS, Oslo, Norway). All variables were weighted by 1/standard deviation. A PCA was also performed on data from each species. For this PCA, all data were used without discriminating between whether livers had been frozen or not. The samples from Tobis (August 2016) were, however, not included in the PCA because they had much higher levels of peroxides and free fatty acids than samples from Karbak.

## 3. Results and Discussion

### 3.1. Principal Component Analysis of All Data

The raw data consist of 278 samples, grouped together according to species and sampling months. Samples from July 2016 were not included because the storage condition of these samples was unknown. To provide a graphical overview of the relationship and patterns in the variation in the data, a principal component analysis (PCA) on all data was performed.

The PCA plot for the data set including all mean values of the livers from the five fish species caught per month. In the scores plot in Figure 1a, clusters of the different species of fish are observed with most monkfish samples located to the left and most coalfish samples located to the right. Overlaps between the different species are observed, indicating similarities between the five species. By comparing Figure 1a with the loadings plot in Figure 1b, it is found that monkfish generally had high FFA content, but lower dry matter and oil content than the other species. The opposite was observed for coalfish. Some coalfish samples were located in the same direction as PV, suggesting that PV was high in these coalfish samples. Based on Figure 1, cod, coalfish, ling and hake were found to generally have higher content of EPA and DHA than monkfish. The different sampling months were located within the inner ellipse of the loadings plot, indicating that different sampling months contributed less to explaining the variation in the data than other variables (measured variables such as PV, FFA, oil, EPA and DHA), which were located between the inner and outer ellipses.

### 3.2. Effect of Storage Conditions

A PCA was also performed to analyse whether storage conditions (on ice (I) or frozen (F)) had any effect on fatty acid composition, FFA or PV for all the fish species. However, no correlation between the measured variables and storage conditions was observed (Figure A1).

To further study whether storage conditions affected individual species, PCA models were constructed for each fish species. It was observed that ling, hake and coalfish had indications of slight effects on storage conditions. For ling specifically, it was observed that the PV was generally higher for samples that had been stored in frozen conditions only (Figure A2). However, the opposite was observed for cod for which fish stored on ice at sea generally had a higher PV (Figure A3). Furthermore, it was observed from the PCA model that the days of storage at sea did not affect the oil composition nor the peroxide value and percentage of free fatty acids in the fish liver. Due to these observations, the effect of storage condition and time at sea was neglected when plotting and examining the data of the individual species and their corresponding PCA plots, as shown in Figure 2.

Figure 2 shows a clear correlation between dry matter and lipid content in all fish species. Such a correlation is expected as oil content contributes to the total dry matter content. This interpretation is further confirmed by the raw data in Table 2, which shows that the patterns in the seasonal variation in dry matter and lipid content are quite similar. Comparison of the values of dry matter and lipid content also shows that the lipids constituted between 72% to 95% of the dry matter content. This was in accordance with a study by Eliassen and Vahl [17], who investigated seasonal variations in water and fat content of cod livers. In the PCA plot in Figure 2, there is a repeating pattern of April 2018 being negatively correlated to dry matter and oil content. This finding suggests that the values of these variables in this specific month were lower compared to samples from other months. The observations regarding low dry matter and lipid content in April 2018 were, in general, confirmed by the raw data in Table 2. However, due to large biological variation differences between values obtained for livers from April 2018 and other sampling months were not significant for all species, but there was a clear trend that both dry matter and lipid content were lowest or second lowest in April 2018 for all species. Furthermore, dry matter content for monkfish was lower compared to the other fish species, but relatively stable throughout the year with a content of 44.1–54.3%. Although the differences were not always significant, ling, coalfish, hake and cod all followed a similar pattern, with a trend to a slight increase in dry matter content observed during autumn and a decrease in spring followed by the previously mentioned low dry matter content in April 2018, of approximately 60%. Moreover, for the lipid content, differences between sampling points were often not significant due to large biological variation, but the general trends were similar to those described for dry matter content. For example, lipid content for ling increased from 55.6% in July 2016 to 63.2% in January 2017, whereafter it decreased to 55.4% in March 2017. Then, it increased significantly to 65.7% in September 2017, whereafter it significantly decreased to approximately 52% in March and April 2018.

The lipid content for monkfish was observed to be the lowest, ranging from content between 31.1% in April 2018 and 43.4% in January 2017, while the other four fish species generally had a lipid content ranging between 42.4–75.9%. The lipid content for coalfish was the highest in six out of nine sampling points and ranged between 48.9% in April 2018 to 75.9% in August 2016.

Røjbæk et al. [9] reported that the lipid content of Baltic cod varied between 50% to 60%, which was in accordance with the levels observed in the present study (49.7–64.3%). Falch et al. [18] compared lipid contents in livers from cod caught in spring, autumn and summer in the Barents Sea (59% to 76%), Icelandic waters (45% to 65%) and southern coast of Ireland (30% to 40%). The livers of the present study, thus, resembled livers from Icelandic waters the most. Icelandic waters are closer to the fishing grounds along the southern part of the Norwegian coast—where the fish in this study were caught—than the other fishing grounds in the study by Falch et al. [18]. They also reported the lipid content in coalfish (saithe) and ling caught from the same fishing grounds, as mentioned above, except that no ling were caught in the Barents Sea. Again, the values obtained for livers from Icelandic waters (46% to 72% for coalfish and 47% to 74% for ling) were more similar to the values obtained in this study (48.9% to 75.9% for coalfish and 51.4% to 65.7% for ling) than the values obtained from other fishing grounds. Falch et al. [18] performed a two-way analysis of variance on the effect of fishing ground and season. They found that the effect of fishing ground was significant for all fish species, whereas season significantly influenced lipid content in ling and coalfish, but not in cod. Dominguez-Petit et al. [19] reported lipid content as varying between 62% to 75% in European hake, which also corresponded well with the observations in our study.

The obvious difference in both dry matter and oil content between monkfish and coalfish, ling, hake and cod may be due to monkfish being from the Lophiiformes order while the other four are from the Gadiformes order of fish. Likewise, according to United States Department of Agriculture, the overall fat content in raw monkfish is found to be 1.52%, while it is 0.67% for cod [20]. This difference and the possibility of monkfish storing its fat differently from cod species (e.g., less fat in the liver) may be the cause for differences observed in the data.

### 3.3. Fatty Acid Composition

The scope of this study is to evaluate the potential of the livers as a source of oils rich in long-chain omega-3 fatty acids. Therefore, the focus will mainly be on the sum of all omega-3 PUFA and omega-6 PUFA as well as EPA and DHA. Nevertheless, the total fatty acid composition of livers from the five fish species from the sampling in November 2016 is shown in Table 3.

Overall, the fatty acid composition of the livers from the different fish species followed the same pattern. The fatty acid present in highest concentration was C18:1 n-9, which constituted between 12.6% (coalfish) and 15.8% (ling) of the total fatty acids. Røjbek et al. [9] also found that C18:1 n-9 was the most prominent fatty acid in cod liver, and it constituted 20.6–24.1% of the fatty acids in triglycerides. DHA was the second most prominent fatty acid and constituted approximately 13% of the total fatty acids. The sum of C20:1 n-9 and C20:1 n-11 was another group of monounsaturated fatty acids present in high amounts (9.8% in monkfish to 12.0% in coalfish). In total, monounsaturated fatty acids (MUFAs) constituted ca. 42% to 44% of the fatty acids, which was slightly higher than the results reported by Falch et al. in livers from saithe (coalfish), ling and cod from Icelandic waters caught in the autumn (35.5–41.0%) [18]. PUFAs constituted between 30% to 31% of the fatty acids, which was slightly lower than found in the study by Falch et al. [18] (32.9–34.8%). The major part of PUFA were omega-3 fatty acids, and the omega-3 to omega-6 ratio varied between 9.2 (monkfish) to 11.6 (coalfish). It has been proposed that in order to have an intake of PUFA that benefits health and prevents disease, the intake of omega-6 PUFA should not be more than two times higher than the omega-3 PUFA intake [21]. This corresponds to an omega-3:omega-6 ratio of >0.5. Therefore, liver oils from these five fish species have a highly health beneficial omega-3:omega-6 ratio.

The total sum of saturated fatty acids was between 19.4% to 21.7%, with C16:0 constituting the major part (from 11.7% in cod to 13.2% in monkfish).

### 3.4. Sum of Omega-6 PUFA

With a few exceptions, Figure 2 did not show strong correlations between sampling months and content of omega-6 PUFA. This was confirmed by the raw data in Table 4, which showed that all fish types had a similar omega-6 PUFA content ranging from 2.35–3.50% until March 2017. Thereafter, the differences between species became larger, with hake and monkfish experiencing a decrease in September 2017 to 1.46% and 1.96%, respectively. The omega-6 PUFA content continued to decrease significantly for monkfish until April 2018. In contrast, the omega-6 PUFA content increased significantly between September 2017 to October 2017 for hake, to 2.86%. Thereafter, the omega-6 content for hake was at the same levels as those of ling, coalfish and cod (2.3–3.6%) for the rest of the sampling period. High standard deviations were observed for some of the data points due to large variation between individual samples of fish.

Røjbek et al. [9] reported that the total content of omega-6 PUFA in cod livers varied during the season from 4.8–5.3% in triacylglycerols and 3.2–4.5% in phospholipids. Méndez [22] reported a seasonal variation of omega-6 PUFA (C18:2 n-6) in hake from 2.0% to 2.8%. McGill and Moffat [23] studied the fatty acid composition in different commercial liver oils. They found that monkfish and coalfish liver oils contained 1.7% and 1.6% omega-6 PUFA (C18:2 n-6). These values were all within the ranges observed for the different species in the current study.

### 3.5. Sum of Omega-3 PUFA in % of Total Fatty Acids

The omega-3 PUFA content in the livers ranged from 19.0% to 29.1% and were relatively stable during the seasons (Table 5). However, a slight decrease was observed in both March 2017 and March 2018 compared to the previous months. This could also be observed in Figure 2, where a negative correlation between omega-3 PUFA content and March 2017 and/or March/April 2018 were observed for all fish species. The decrease was, however, only significant for coalfish and cod and only in March 2018. Overall, cod liver was found to have the highest values of omega-3 PUFA in the sampling period. Thus, cod liver had the highest or second highest omega-3 PUFA content at eight out of the nine sampling points. The omega-3 PUFA content reached the lowest levels in ling and coalfish (only approximately 20%). These low levels of omega-3 PUFA were found towards the end of the sampling period.

Røjbek et al. [9] found that omega-3 PUFA levels in cod livers from Baltic Sea varied during the season from 35–42% in triacylglycerols and from 46% to 51% in phospholipids. These values were substantially higher than those found in our study. The fatty acid composition in livers varies with the diet of the fish, which may be different between cods caught in the Baltic Sea and in the North Sea. This could explain the large difference in the omega-3 PUFA content between the two studies. Variations in diet, most likely, also explained the minor fluctuations in the omega-3 PUFA content observed in March in both 2017 and 2018. Moreover, seasonal variations are also due to the reproductive cycle, at least in female species [9]. It is known that these five fish species generally have spawning season around spring (ranging between January and May) [24]. Spawning season has, in previous studies, been shown to cause reduced muscle tissue in favor of gonad maturation in farmed Atlantic cod [25,26]. The growth of fish is normalized post-spawning season (around October). In gadoid species such as cod, the liver is the primary energy reserve as the liver lipids are mobilized when more energy is required [27]. It is a possibility that the extra energy used before and during spawning season results in lower lipid content in the liver, meaning less oil in the liver will be present for extraction. However, the livers in this study are both from females and males, so the effect of the spawning season might be lower than if the livers had only been from females. Nevertheless, results in Table 2 show lower lipid content in the samples for April and March, which may support the mentioned hypothesis. To fully investigate a correlation between seasons (including spawning period) and liver oil attributes, more data are required. This means that more data from the summer seasons and, generally, more data from the same seasons in different years would be beneficial.

### 3.6. Content of EPA in % of Total Fatty Acids

The livers from all five species of fish were, in general, found to follow a similar pattern, with respect to their EPA as that observed for the omega-3 PUFA content in Figure 2; this means that levels of EPA were, in general, stable throughout the seasons, but with the lowest levels in March 2017 and/or March/April 2018. However, Table 6 showed that EPA levels were not significantly lower at these sampling points than at most other sampling points. Table 6 also confirmed that cod liver had a relatively stable EPA content, which ranged between 5.6% to 8.6%, and it generally contained the highest amount of EPA compared to the other species (at seven out of the nine sampling points). Falch et al. [18] found that EPA content in cod livers from Icelandic waters varied between 10.1% in the spring to 8.5% in the autumn. In our study, the EPA content in cod livers tended to be highest in the autumn and lowest in the early spring. Monkfish liver was found to have the lowest EPA content, which agreed with the interpretation from the PCA in Figure 1. The EPA content in monkfish livers varied between 4.6% to 6.4%. McGill and Moffat [23] observed an EPA content of 7.7% in commercial monkfish liver oil, which is a little higher than observed in our study.

The EPA content in livers from ling and hake varied between the levels observed for livers from monkfish and cod. Falch et al. [18] found that the EPA content in ling livers from Icelandic waters was lowest in the autumn (4.8%) and highest in the spring (6.3%). Again, this was in contrast to the pattern observed in our study, where ling livers tended to have the highest values in the early autumn. We found slightly higher levels of EPA in our study. EPA content in hake livers from Argentina–Uruguay varied from 5.0% to 8.4%, with the highest values observed in late summer and the lowest in winter [22]. In our study, there was no clear seasonal effect on the EPA content of hake livers.

Coalfish livers had a large variation in their EPA content (4.4% to 9.0%). Although a similar large variation was not observed for coalfish livers caught in Icelandic waters (6.3% to 9.0%), the EPA levels were almost in the same range.

### 3.7. Content of DHA in Livers as % of Total Fatty Acids

The values of DHA in the livers from the five species generally ranged between 10.3% to 14.6% DHA (Table 7). In Figure 2, the pattern of DHA was almost similar to that of EPA for ling and monkfish. The location of DHA in the loadings plots for these two species suggested that DHA levels were low in March 2017 (monkfish) or April 2018 (ling). This was confirmed by the results in Table 7. However, DHA content was not significantly different between any sampling points for these two species. High standard deviations for monkfish and ling were observed in March 2017, meaning that the low DHA % in this month may be caused by a random and unexplainable variation within samples of the two species in that month.

Figure 2 also showed that for the other three species, DHA levels behaved differently from EPA levels. For hake and coalfish, high levels of DHA were found in January 2017 (14.6%) and March 2017 (13.3%), respectively, whereas for cod a high level was observed in April 2018 (14.4%). For these species, the DHA levels were significantly lower in July and August 2016 for coalfish (10.5% and 10.8%, respectively), October 2017 (11.7%) for hake and in August 2016 (11.2%) for cod compared to the months mentioned above for the high DHA levels for the corresponding fish species.

The highest values were, in general, observed for cod livers (five out of the nine sampling times). The observations in Figure 1, that monkfish livers had lower DHA content than livers from the other fish species, were only confirmed by the data in Table 7 in March 2017 and March 2018.

The study by Falch et al. [18] reported lower DHA values for livers from cod (6.0% to 9.7%), ling (5.4% to 9.4%) and coalfish (6.2% to 9.4%) when these species were caught in Icelandic waters compared to the findings in this study. In contrast, livers from hake caught in waters of Argentina–Uruguay had slightly higher levels of DHA (12.7–17.6%) than found in our study [22]. Moreover, McGill and Moffat [23] found DHA levels of 12.4% and 14.2% in commercial coalfish and monkfish liver oils. These values are within the ranges found in the present study. Differences between fatty acid compositions reported in the different studies may be due to the different catching grounds and, thereby, the different diets of the fish.

### 3.8. Liver Oils as a New Source of Fish Oils

Moving the focus back to the possibilities of utilizing the fishing waste from the five species of fish in the production of marine oil for human consumption, the recommended values of EPA and DHA in fish liver oil are 7–16% and 6–18%, respectively [28]. The most suitable species for the production of liver oil are, therefore, cod and hake, as they generally had a high content of both EPA and DHA. Coalfish livers were also observed to have high contents of EPA and DHA. However, the livers from this species seemed to vary more during the seasons, and the EPA content dropped to ca. 5% and below in some months. This may make coalfish less suitable for all-year production than cod and hake. Ling livers only had higher EPA levels than recommended for two of the sampling months, but DHA levels were within the recommended levels in all sampling months. Monkfish livers had lower EPA levels than recommended in all sampling months, but DHA levels were within the recommended levels. Since monkfish livers had the lowest lipid content throughout the sampling period, this was the least suitable fish for liver oil production.

In a high-quality fish oil ready for human consumption, the omega-6 PUFA content should be as low as possible to combat the average consumer’s generally higher intake of omega-6 PUFA compared to omega-3 PUFA. In a typical cod liver oil, the total omega-6 fatty acid content is 3.6% while the omega-3 content is approximately 24% [29]. A similar ratio between omega-6 and omega-3 PUFA was observed for all fish species throughout the sampling period. This showed that the oils in the fish livers have great potential for production of a fish oil, with a quality similar to cod liver oil based on the ratio between omega-6 and omega-3 PUFA.

### 3.9. Quality of the Liver

#### 3.9.1. Peroxide Value (PV)

Table 8 shows that from July 2016 to March 2017 the PVs in the liver from the various fish were similar (<1 meq/kg), with the exception of samples from August 2016, which were from another fishing vessel than the rest of the samples. Then, an increase in PV was observed for coalfish, ling and cod in September or October 2017. However, the increase in PV was only significant for cod. Coalfish liver was found to have the highest value in October 2017 of 2.8 meq/kg, while livers from hake and monkfish still maintained a PV under 1 meq/kg throughout these months. It is important to note that the standard deviations for the coalfish and cod samples were observed to be high for the months September 2017, October 2017 and March 2018, indicating that the data points were spread out over a wider range of values and were generally not very consistent. When examining the raw data, the storage condition was observed not to correlate well with the observed deviations, which may indicate that variation between individual samples was the reason for the variation in data.

#### 3.9.2. Free Fatty Acid (FFA)

The FFA content generally differed for the various fish species, with livers from hake and ling sharing the same pattern throughout the seasons (Table 8). Monkfish livers had the highest values of FFA, with a tendency of higher levels in March 2017 and September 2017, with values of approximately 6% FFA.

#### 3.9.3. Overall Quality

The overall quality of fish oil is dependent on the quality of raw materials (fish liver) and the processing that takes place. In marine products, avoidance of oxidation and rancidity is the primary focus point. PV of the liver may increase during processing, and PV should, therefore, be as low as possible in the raw materials. Fish livers contain endogenous lipase, which could hydrolyse triacylglycerols and phopholipids as well as release free fatty acids. This may happen both during processing and storage, particularly at room temperature and higher. Preferably, the final fish oil product should have low values of PV and FFA %, indicating low oxidation and higher stability of the product. Crude fish oil intended for human consumption generally has a peroxide value ranging between 3–20 meq/kg [30]; the lower the value is, the better the quality of the final refined fish oil. For the FFA content, the recommended value for crude marine oils is 1–7%, however, it is more commonly found to be between 2–5% [30].

As observed in Table 8, the obtained FFA % for all fish liver types was found to be <7%, which indicates that they are suitable for human consumption. Even if the fish liver may experience further oxidation and hydrolysis that will lower the quality during the oil extraction process, the extracted crude oil will undergo further refining and deodorization steps that will remove free fatty acids, lipid hydroperoxides and volatile decomposition productions from the oils. It is mainly the volatile oxidation products that are responsible for undesirable fishy and rancid off-flavors. Nevertheless, monkfish with a FFA content close to 7% may be less suitable for oil production than the other fish species. None of the fish livers were found to have too high peroxide values, which makes them suitable for producing fish oil for human consumption.

## 4. Conclusions

All fish species had similar tendencies of generally lower oil content in the livers in spring and higher content in the fall. This could be due to spawning season, diet or species variations. For EPA and DHA, the values recommended for fish liver oils are 7–16% and 6–18%, respectively. For PV and FFA, values of crude oils should be as low as possible but are usually within 3–20 meq/kg and 1–7%, respectively. Considering these recommendations, cod, hake and, to some extent, ling had EPA and DHA levels, PV and FFA contents that are within the recommended levels for fish oil for human consumption. Livers from monkfish may be less suitable for the production of fish oil for human consumption due to their lower content of oil, lower levels of EPA and higher FFA content. Coalfish could potentially be used for fish liver oil production. However, the fluctuating EPA content and PV in coalfish livers may make it difficult to achieve a standardised production of oil. Thus, livers from cod, hake and ling have the highest potential as a new source of omega-3 PUFA for human consumption. They all have an omega-3: omega-6 PUFA ratio above 9.5, which is well above the suggested ratio of >0.5 for a healthy diet.

## Figures and Tables

**Figure 1 foods-11-00905-f001:**
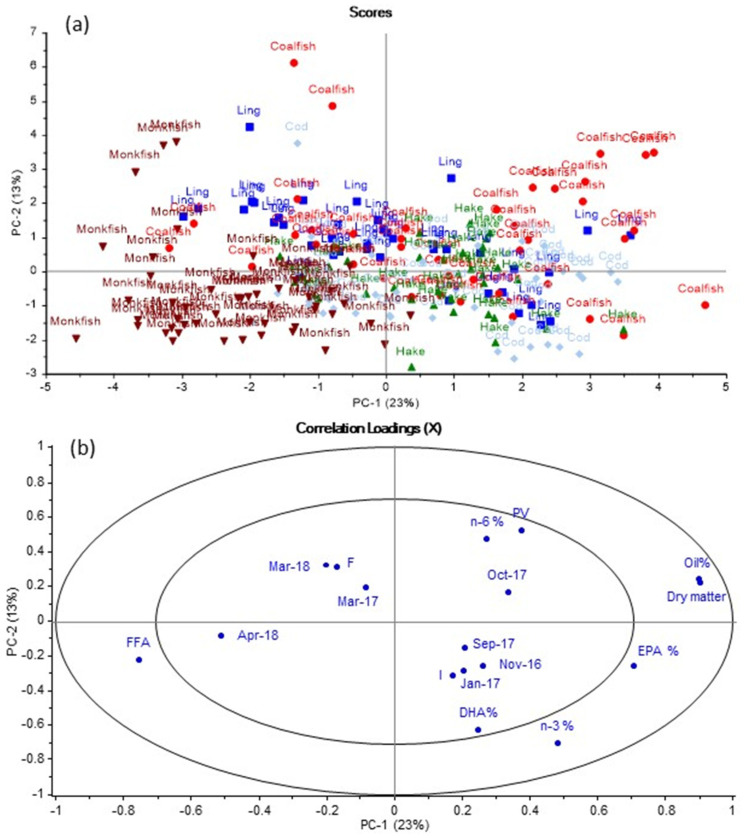
PCA plot for all fish species per month and the collected data for dry matter content, oil content, fatty acid composition, peroxide value and free fatty acid content. Data for July 2016 are not included in the PCA plots. (**a**) Scores plot of all the different samples and (**b**) loadings plot of variables including selected fatty acids, month of sampling and quality parameters. Variables located within the inner ellipse explain less than 50% of the variation in the data. Variables located between the two ellipses explain between 50% and 100% of the variation in the data.

**Figure 2 foods-11-00905-f002:**
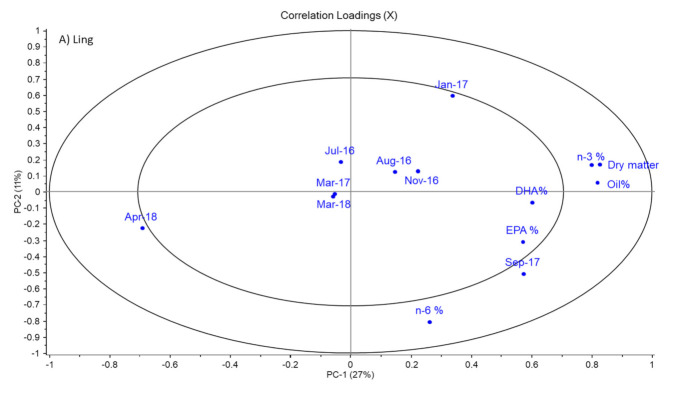
Individual PCA plots for all fish species, including specific sampling months and the data collected for dry matter content, oil content and fatty acid composition (n-6, n-3, EPA, DHA). Peroxide value and free fatty acids. Storage conditions are not included. Variables located within the inner ellipse explain less than 50% of the variation in the data. Variables located between the two ellipses explain between 50% and 100% of the variation in the data.

**Table 1 foods-11-00905-t001:** Number of samples received at different sampling times for each species and treatment.

Species	Hake		Cod		Ling		Coalfish		Monkfish	
Storage	Ice	Frozen	Ice	Frozen	Ice	Frozen	Ice	Frozen	Ice	Frozen
Jul-16 †	3	3	3	3	3	3	2	2	3	3
Aug-16 *	5	5	3	5	1	0	3	5	3	5
Nov-16	2	0	2	0	2	0	2	0	2	0
Jan-17	2	1	3	3	3	2	3	3	3	2
Mar-17	2	2	2	2	4	4	4	4	4	4
Sep-17	2	3	3	2	3	3	3	2	3	3
Oct-17	5	5	6	6	0	0	5	4	6	5
Mar-18			3	3	2	3	3	2	3	2
Apr-18	6	4	7	7	7	7	3	3	7	7
Sum of each species within a treatment	27	23	32	31	25	22	28	25	34	31
Sum of each species	50		63		47		53		65	

† These samples was marked with 1 and 2 and not frozen and ice, however, similar numbers were sampled for ice and frozen storage within each spicies. Despite the lack of identification the samples has been included in the interpretation of results after knowing that storage before landing had no effect. * The samples from this sampling point was from Tobis. All other samples (sampling points) were from Karbak.

**Table 2 foods-11-00905-t002:** Dry matter and oil contents in fish livers sampled at different time points.

**Dry Matter [%]**					
**Sampling**	**Ling**	**Coalfish**	**Hake**	**Cod**	**Monkfish**
Jul 2016	65.8 ± 3.4 ^a,b,c^	79.1 ± 3.3 ^c^	55.7 ± 4.6 ^a^	71.2 ± 4.2 ^c^	48.7 ± 1.9 ^a,b^
Aug 2016	70.4 ± 0.0 ^a,b,c^	82.6 ± 3.0 ^c^	62.4 ± 10.3 ^a,b,c^	65.7 ± 9.2 ^a,b,c^	49.3 ± 12 ^a,b^
Nov 2016	71.2 ± 2.4 ^a,b,c^	82.3 ± 1.1 ^c^	76.5 ± 7.9 ^c^	70.2 ± 0.8 ^a,b,c^	50.9 ± 0.0 ^a,b^
Jan 2017	71.6 ± 1.6 ^b,c^	75.3 ± 5.1 ^b,c^	62.3 ± 8.3 ^a,b,c^	68.0 ± 4.6 ^b,c^	54.3 ± 4.6 ^b^
Mar 2017	65.5 ± 4.0 ^a,b,c^	68.4 ± 4.0 ^a,b^	69.6 ± 1.8 ^b,c^	65.5 ± 5.8 ^a,b,c^	48.6 ± 6.3 ^a,b^
Sep 2017	73.2 ± 2.2 ^c^	78.3 ± 1.7 ^c^	67.6 ± 2.7 ^b,c^	63.9 ± 6.7 ^a,b,c^	49.9 ± 5.4 ^a,b^
Oct 2017	Nd	80.4 ± 3.5 ^c^	66.3 ± 2.8 ^b,c^	72.7 ± 3.6 ^c^	50.9 ± 3.4 ^a,b^
Mar 2018	62.4 ± 6.3 ^a,b^	61.2 ± 3.1 ^a^	Nd	60.3 ± 2.8 ^a,b^	44.7 ± 2.1 ^a,b^
Apr 2018	59.0 ± 7.6 ^a^	61.7 ± 7.6 ^a^	60.6 ± 4.4 ^a,b^	58.0 ± 5.8 ^a^	44.1 ± 3.3 ^a^
**Oil [%]**					
**Sampling**	**Ling**	**Coalfish**	**Hake**	**Cod**	**Monkfish**
Jul 2016	55.6 ± 4.5 ^a,b,c^	73.3 ± 1.4 ^c,d^	42.4 ± 5.0 ^a^	62.4 ± 3.9 ^b,c^	37.5 ± 2.2 ^a,b^
Aug 2016	61.4 ± 0.0 ^a,b,c^	75.9 ± 4.6 ^d^	51.5 ± 13 ^a,b^	53.2 ± 13 ^a,b^	38.5 ± 14 ^a,b^
Nov 2016	59.4 ± 0.8 ^a,b,c^	72.0 ± 2.6 ^c,d^	66.6 ± 12 ^b^	59.1 ± 0.3 ^a,b,c^	40.9 ± 0.0 ^a,b^
Jan 2017	63.2 ± 3.3 ^b,c^	65.8 ± 5.3 ^b,c^	51.2 ± 9.8 ^a,b^	58.4 ± 5.7 ^a,b,c^	43.4 ± 7.2 ^b^
Mar 2017	55.4 ± 6.0 ^a,b^	60.2 ± 4.4 ^b^	61.5 ± 3.7 ^b^	56.7 ± 7.8 ^a,b,c^	37.6 ± 7.9 ^a,b^
Sep 2017	65.7 ± 2.8 ^c^	70.8 ± 2.9 ^c,d^	58.2 ± 3.0 ^b^	52.6 ± 8.7 ^a,b,c^	38.9 ± 6.1 ^a,b^
Oct 2017	Nd	73.2 ± 4.1 ^c,d^	57.0 ± 3.3 ^b^	64.3 ± 4.3 ^c^	39.4 ± 4.8 ^a,b^
Mar 2018	51.4 ± 7.6 ^a,b^	51.2 ± 3.8 ^a^	Nd	49.7 ± 4.5 ^a,b^	32.9 ± 2.9 ^a,b^
Apr 2018	52.2 ± 7.2 ^a^	48.9 ± 7.2 ^a^	49.2 ± 5.8 ^a,b^	50.3 ± 5.5 ^a^	31.1 ± 4.5 ^a^

Refer to Table 1 for the number of livers sampled for each species at each time point. Values in the same column with the same letter a, b… are not significantly different. Nd is not determined.

**Table 3 foods-11-00905-t003:** Fatty acids (% of total fatty acids) measured in the extracted oil from livers of different fish species sampled in November 2016. Only fatty acids present in concentrations >0.1% are shown.

	Ling	Coalfish	Hake	Cod	Monkfish
C14:0	4.11 ± 0.4	4.91 ± 0.4	5.95 ± 1.6	4.23 ± 0.3	4.25 ± 0.4
C14:1	0.19 ± 0.0	0.20 ± 0.0	0.24 ± 0.0	0.22 ± 0.0	0.18 ± 0.0
C15:0	0.40 ± 0.0	0.40 ± 0.0	0.48 ± 0.1	0.43 ± 0.0	0.46 ± 0.0
C16:0	12.1 ± 0.0	12.0 ± 0.3	12.8 ± 1.0	11.7 ± 0.2	13.2 ± 0.3
C16:1 (n-7)	4.18 ± 0.2	4.45 ± 0.4	4.68 ± 0.3	4.75 ± 0.0	6.36 ± 0.0
C16:2 (n-4)	0.24 ± 0.1	0.56 ± 0.0	0.46 ± 0.1	0.37 ± 0.0	0.42 ± 0.0
C16:3 (n-4)	0.32 ± 0.0	0.34 ± 0.0	0.34 ± 0.0	0.39 ± 0.0	0.33 ± 0.0
C17:0	0.14 ± 0.0	0.36 ± 0.0	0.25 ± 0.2	0.21 ± 0.0	0.31 ± 0.1
C16:4 (n-3)	0.21 ± 0.0	0.58 ± 0.1	0.39 ± 0.3	0.31 ± 0.0	0.39 ± 0.0
C18:0	2.61 ± 0.1	2.86 ± 0.3	2.21 ± 0.9	2.85 ± 0.0	2.64 ± 0.2
C18:1 (n-9)	15.8 ± 1.6	12.6 ± 0.8	12.4 ± 0.3	14.1 ± 0.1	12.3 ± 0.1
C18:1 (n-7)	2.90 ± 0.3	2.26 ± 0.0	2.19 ± 0.2	3.13 ± 0.1	3.15 ± 0.1
C18:2 (n-6)	1.63 ± 0.1	1.48 ± 0.0	1.67 ± 0.1	1.32 ± 0.1	1.64 ± 0.3
C18:2 (n-4)	0.17 ± 0.0	0.23 ± 0.0	0.18 ± 0.1	0.19 ± 0.0	0.19 ± 0.0
C18:3 (n-6)	0.12 ± 0.0	0.14 ± 0.0	0.13 ± 0.0	0.11 ± 0.0	0.13 ± 0.0
C18:3 (n-4)	0.14 ± 0.0	0.12 ± 0.0	0.13 ± 0.0	0.14 ± 0.0	0.15 ± 0.0
C18:3 (n-3)	1.16 ± 0.1	1.24 ± 0.1	1.37 ± 0.1	1.04 ± 0.0	1.13 ± 0.2
C18:4 (n-3)	2.51 ± 0.2	2.45 ± 1.3	3.23 ± 0.6	2.43 ± 0.1	2.89 ± 0.4
C20:1 (n-9, n-11)	10.9 ± 0.4	12.0 ± 0.8	11.2 ± 0.8	10.0 ± 0.3	9.77 ± 0.7
C20:1 (n-7)	0.31 ± 0.0	0.19 ± 0.0	0.24 ± 0.0	0.44 ± 0.0	0.34 ± 0.0
C20:2 (n-6)	0.34 ± 0.0	0.27 ± 0.0	0.31 ± 0.0	0.39 ± 0.0	0.29 ± 0.0
C20:4 (n-6)	0.66 ± 0.0	0.45 ± 0.0	0.54 ± 0.1	0.83 ± 0.1	0.76 ± 0.0
C20:3 (n-3)	0.24 ± 0.0	0.17 ± 0.0	0.22 ± 0.0	0.24 ± 0.0	0.21 ± 0.0
C20:4 (n-3)	0.93 ± 0.0	0.89 ± 0.1	0.95 ± 0.1	0.86 ± 0.0	0.85 ± 0.1
C20:5 (n-3)	7.05 ± 0.1	7.85 ± 0.2	7.72 ± 0.5	8.13 ± 0.2	6.21 ± 0.3
C22:1 (n-11)	9.41 ± 0.6	9.51 ± 0.1	9.51 ± 0.7	8.70 ± 0.2	9.19 ± 0.2
C22:1 (n-9)	0.55 ± 0.0	0.52 ± 0.0	0.59 ± 0.1	0.45 ± 0.0	0.72 ± 0.0
C21:5 (n-3)	0.45 ± 0.0	0.54 ± 0.1	0.49 ± 0.1	0.44 ± 0.1	0.39 ± 0.0
C22:5 (n-3)	1.50 ± 0.1	1.54 ± 0.0	1.40 ± 0.1	1.85 ± 0.1	1.60 ± 0.0
C22:6 (n-3)	12.9 ± 0.1	12.8 ± 0.2	12.8 ± 0.4	13.3 ± 0.0	13.1 ± 0.0
C24:1 (n-9)	0.57 ± 0.1	0.55 ± 0.0	0.58 ± 0.1	0.54 ± 0.1	0.67 ± 0.1
Total sum	94.9 ± 0.1	94.6 ± 1.1	95.8 ± 0.2	94.3 ± 0.6	94.5 ± 0.2
Sum of omega-3 PUFA	27.0 ± 0.0	28.0 ± 1.8	28.5 ± 1.4	28.6 ± 0.3	26.8 ± 0.4
Sum of omega-6 PUFA	2.85 ± 0.0	2.41 ± 0.0	2.72 ± 0.2	2.72 ± 0.0	2.91 ± 0.3
Omega-3:omega-6 ratio	9.5	11.6	10.5	10.5	9.2

**Table 4 foods-11-00905-t004:** Sum of omega-6 PUFA (%) in fish livers sampled at different time points.

Sampling	Ling	Coalfish	Hake	Cod	Monkfish
Jul 2016	2.88 ± 0.25 ^a^	2.35 ± 0.03 ^a^	3.11 ± 0.16 ^b,c^	2.55 ± 0.11 ^a^	2.94 ± 0.13 ^b,c^
Aug 2016	2.59 ± 0.00 ^a^	2.38 ± 0.11 ^a^	2.91 ± 0.23 ^b,c^	2.99 ± 0.66 ^a,b^	3.04 ± 0.26 ^c^
Nov 2016	2.85 ± 0.04 ^a^	2.41 ± 0.04 ^a,b^	2.72 ± 0.23 ^b,c^	2.72 ± 0.04 ^a,b^	2.71 ± 0.00 ^a,b,c^
Jan 2017	2.54 ± 0.29 ^a^	2.53 ± 0.48 ^a^	2.92 ± 0.11 ^b,c^	2.76 ± 0.35 ^a,b^	2.91 ± 0.19 ^b,c^
Mar 2017	3.24 ± 1.15 ^a^	2.49 ± 0.21 ^a^	2.99 ± 0.05 ^b,c^	3.01 ± 0.13 ^a,b^	3.50 ± 1.54 ^c^
Sep 2017	3.91 ± 1.95 ^a^	3.33 ± 0.31 ^b,c^	1.46 ± 0.12 ^a^	3.28 ± 0.28 ^b^	1.96 ± 0.49 ^a,b^
Oct 2017	Nd	3.54 ± 0.24 ^c^	2.86 ± 0.09 ^b^	2.76 ± 0.29 ^a,b^	1.85 ± 0.19 ^a^
Mar 2018	3.01 ± 0.23 ^a^	3.59 ± 0.66 ^c^	Nd	2.84 ± 0.22 ^a,b^	1.63 ± 0.17 ^a^
Apr 2018	2.30 ± 0.27 ^a^	2.77 ± 0.47 ^a,b^	3.09 ± 0.16 ^c^	2.76 ± 0.15 ^a,b^	1.86 ± 0.23 ^a^

Refer to Table 1 for the number of livers sampled for each species at each time point. Values in the same column with the same letter a, b… are not significantly different. Nd is not determined.

**Table 5 foods-11-00905-t005:** Sum of omega-3 PUFA (%) in fish livers sampled at different time points.

Sampling	Ling	Coalfish	Hake	Cod	Monkfish
Jul 2016	23.2 ± 1.1 ^a,b^	27.3 ± 0.7 ^c^	24.8 ± 0.8 ^a^	28.4 ± 1.5 ^c^	23.9 ± 0.8 ^a,b^
Aug 2016	26.6 ± 0.0 ^a,b^	27.3 ± 1.8 ^c^	27.9 ± 2.9 ^b^	27.0 ± 2.9 ^b,c^	25.2 ± 2.1 ^a,b^
Nov 2016	27.0 ± 0.0 ^b^	28.0 ± 1.8 ^c^	28.5 ± 1.4 ^a,b^	28.6 ± 0.3 ^b,c^	26.5 ± 0.0 ^a,b^
Jan 2017	25.8 ± 1.6 ^b^	26.1 ± 1.0 ^c^	27.4 ± 1.3 ^a,b^	29.1 ± 0.9 ^c^	25.8 ± 0.9 ^b^
Mar 2017	22.9 ± 2.0 ^a,b^	25.5 ± 1.2 ^c^	26.6 ± 0.6 ^a,b^	26.5 ± 1.6 ^b,c^	23.4 ± 3.3 ^a,b^
Sep 2017	26.9 ± 5.2 ^b^	25.6 ± 3.0 ^c^	27.4 ± 1.0 ^a,b^	27.2 ± 2.1 ^b,c^	24.8 ± 0.9 ^a,b^
Oct 2017	Nd	22.2 ± 0.9 ^b^	25.3 ± 1.2 ^a^	27.8 ± 1.0 ^c^	25.1 ± 0.7 ^a,b^
Mar 2018	23.1 ± 0.6 ^a,b^	19.0 ± 2.7 ^a^	Nd	22.8 ± 2.5 ^a^	21.8 ± 3.1 ^a^
Apr 2018	20.0 ± 2.3 ^a^	20.3 ± 1.7 ^a,b^	24.9 ± 1.1 ^a^	25.3 ± 1.3 ^a,b^	24.5 ± 1.8 ^a,b^

Refer to Table 1 for the number of livers sampled for each species at each time point. Values in the same column with the same letter a, b… are not significantly different. Nd is not determined.

**Table 6 foods-11-00905-t006:** EPA content (%) in fish livers sampled at different time points.

EPA [%]					
Sampling	Ling	Coalfish	Hake	Cod	Monkfish
Jul 2016	6.3 ± 0.6 ^a^	7.6 ± 0.5 ^b,c,d^	5.5 ± 0.4 ^a^	7.8 ± 1.0 ^b,c,d^	5.6 ± 0.6 ^a,b^
Aug 2016	7.7 ± 0.0 ^a^	8.2 ± 0.9 ^c,d^	7.5 ± 2.0 ^c^	8.6 ± 1.5 ^d^	6.3 ± 1.2 ^b^
Nov 2016	7.1 ± 0.1 ^a^	7.9 ± 0.2 ^b,c,d^	7.7 ± 0.5 ^a,b,c^	8.1 ± 0.2 ^a,b,c,d^	6.4 ± 0.0 ^a,b^
Jan 2017	6.4 ± 0.7 ^a^	6.0 ± 0.7 ^a,b^	6.2 ± 0.4 ^a,b,c^	7.7 ± 0.3 ^b,c,d^	6.0 ± 0.3 ^a,b^
Mar 2017	5.4 ± 0.6 ^a^	5.2 ± 0.9 ^a^	6.5 ± 0.1 ^a,b,c^	6.8 ± 0.6 ^a,b,c^	4.6 ± 1.6 ^a^
Sep 2017	6.9 ± 3.4 ^a^	9.0 ± 1.0 ^d^	7.3 ± 0.5 ^a,b,c^	8.6 ± 1.1 ^c,d^	5.6 ± 0.4 ^a,b^
Oct 2017	Nd	7.4 ± 0.6 ^b,c^	7.2 ± 0.3 ^b,c^	8.4 ± 0.7 ^d^	5.9 ± 0.5 ^a,b^
Mar 2018	5.7 ± 0.3 ^a^	4.4 ± 0.9 ^a^	Nd	5.6 ± 0.8 ^a^	4.6 ± 1.0 ^a,b^
Apr 2018	5.2 ± 1.0 ^a^	5.3 ± 1.5 ^a^	5.9 ± 0.4 ^a,b^	6.7 ± 0.6 ^a,b^	5.7 ± 0.9 ^a,b^

Refer to Table 1 for the number of livers sampled for each species at each time point. Values in the same column with the same letter a, b… are not significantly different. Nd is not determined.

**Table 7 foods-11-00905-t007:** DHA content (%) in fish livers sampled at different time points.

DHA [%]					
Sampling	Ling	Coalfish	Hake	Cod	Monkfish
Jul 2016	10.3 ± 0.4 ^a^	10.5 ± 0.2 ^a^	12.5 ± 0.7 ^a,b^	12.3 ± 1.0 ^a,b^	11.4 ± 0.9 ^a^
Aug 2016	11.8 ± 0.0 ^a^	10.8 ± 1.0 ^a^	12.6 ± 1.3 ^a,b^	11.2 ± 1.2 ^a^	11.9 ± 1.0 ^a^
Nov 2016	12.9 ± 0.1 ^a^	12.8 ± 0.2 ^a,b^	12.8 ± 0.4 ^a,b,c^	13.3 ± 0.0 ^a,b,c^	13.1 ± 0.0 ^a^
Jan 2017	12.5 ± 0.8 ^a^	13.1 ± 0.9 ^b^	14.6 ± 1.6 ^c^	13.9 ± 0.9 ^b,c^	12.8 ± 1.1 ^a^
Mar 2017	11.0 ± 3.5 ^a^	13.3 ± 0.9 ^b^	13.3 ± 0.6 ^a,b,c^	12.8 ± 1.1 ^a,b,c^	10.8 ± 4.3 ^a^
Sep 2017	12.5 ± 1.2 ^a^	13.1 ± 1.8 ^b^	12.8 ± 0.9 ^a,b,c^	13.2 ± 1.1 ^b,c^	12.9 ± 1.0 ^a^
Oct 2017	Nd	11.6 ± 0.6 ^a,b^	11.7 ± 0.6 ^a^	13.8 ± 0.7 ^b,c^	12.6 ± 0.7 ^a^
Mar 2018	12.8 ± 0.3 ^a^	11.8 ± 1.9 ^a,b^	Nd	13.3 ± 1.4 ^b,c^	11.6 ± 1.9 ^a^
Apr 2018	10.4 ± 1.2 ^a^	11.7 ± 0.9 ^a,b^	13.0 ± 0.5 ^b,c^	14.4 ± 0.9 ^c^	12.5 ± 1.1 ^a^

Refer to Table 1 for the number of livers sampled for each species at each time point. Values in the same column with the same letter a, b… are not significantly different. Nd is not determined.

**Table 8 foods-11-00905-t008:** Peroxide values and free fatty acid content in fish livers sampled at different time points.

**Peroxide Value [meq. ROOH/kg Oil]**					
**Sampling**	**Ling**	**Coalfish**	**Hake**	**Cod**	**Monkfish**
Jul 2016	0.65 ± 0.27 ^a^	0.75 ± 0.49 ^a^	0.36 ± 0.15 ^a^	0.45 ± 0.17 ^a^	0.30 ± 0.25 ^a^
Aug 2016	1.34 ± 0.00 ^a^	3.27 ± 2.88 ^a^	1.31 ± 1.53 ^a^	1.13 ± 1.01 ^ab^	2.89 ± 2.54 ^b^
Nov 2016	0.30 ± 0.04 ^a^	0.43 ± 0.10 ^a^	0.40 ± 0.16 ^a^	0.28 ± 0.11 ^a^	0.66 ± 0.00 ^ab^
Jan 2017	0.47 ± 0.21 ^a^	0.64 ± 0.34 ^a^	0.16 ± 0.14 ^a^	0.30 ± 0.08 ^a^	0.22 ± 0.18 ^a^
Mar 2017	0.62 ± 0.10 ^a^	0.89 ± 0.48 ^a^	0.19 ± 0.03 ^a^	0.73 ± 0.32 ^ab^	0.35 ± 0.11 ^a^
Sep 2017	0.93 ± 0.42 ^a^	2.55 ± 1.11 ^a^	0.34 ± 0.15 ^a^	0.57 ± 0.24 ^a^	0.31 ± 0.14 ^a^
Oct 2017	Nd	2.80 ± 1.81 ^a^	0.49 ± 0.19 ^a^	1.72 ± 0.67 ^b^	0.57 ± 0.20 ^a^
Mar 2018	0.66 ± 0.25 ^a^	2.09 ± 1.78 ^a^	Nd	1.24 ± 0.62 ^ab^	0.91 ± 0.85 ^a^
Apr 2018	0.63 ± 0.38 ^a^	0.94 ± 0.71 ^a^	0.30 ± 0.12 ^a^	0.67 ± 0.23 ^a^	0.25 ± 0.17 ^a^
**FFA [%]**					
**Sampling**	**Ling**	**Coalfish**	**Hake**	**Cod**	**Monkfish**
Jul 2016	1.24 ± 0.46 ^ab^	1.82 ± 0.70 ^a^	2.26 ± 0.28 ^a^	1.74 ± 0.85 ^a^	4.78 ± 0.41 ^a^
Aug 2016	0.67 ± 0.00 ^abc^	1.96 ± 0.74 ^a^	2.05 ± 2.40 ^a^	2.29 ± 2.13 ^a^	4.77 ± 3.37 ^a^
Nov 2016	0.38 ± 0.06 ^a^	1.59 ± 0.02 ^a^	1.27 ± 0.90 ^a^	1.42 ± 0.04 ^a^	3.51 ± 0.00 ^a^
Jan 2017	0.85 ± 0.41 ^a^	2.26 ± 0.59 ^a^	1.24 ± 0.54 ^a^	1.65 ± 0.31 ^a^	4.56 ± 1.43 ^a^
Mar 2017	2.17 ± 0.50 ^bc^	3.27 ± 0.54 ^ab^	1.61 ± 0.33 ^a^	2.26 ± 1.02 ^ab^	5.93 ± 1.33 ^a^
Sep 2017	1.36 ± 0.55 ^ab^	2.15 ± 0.98 ^a^	1.83 ± 0.22 ^a^	4.17 ± 0.89 ^b^	6.23 ± 1.56 ^a^
Oct 2017	Nd	2.10 ± 0.41 ^a^	1.16 ± 0.27 ^a^	2.10 ± 0.47 ^a^	5.06 ± 0.98 ^a^
Mar 2018	1.62 ± 1.07 ^abc^	3.24 ± 1.12 ^ab^	Nd	3.06 ± 0.41 ^ab^	3.57 ± 1.57 ^a^
Apr 2018		4.59 ± 1.68 ^b^	2.50 ± 0.62 ^a^	4.08 ± 0.84 ^b^	5.93 ± 1.15 ^a^

Refer to Table 1 for the number of livers sampled for each species at each time point. Values in the same column with the same letter a, b… are not significantly different. Nd is not determined.

## Data Availability

No new data were created or analyzed in this study. Data sharing is not applicable to this article.

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
