# Peer review of "Fish Liver Discards as a Source of Long-Chain Omega-3 Polyunsaturated Fatty Acids"

_foods, 2022, doi:10.3390/foods11070905_

Round 1
Reviewer 1 Report
This research evaluate the potential of using livers from different fish species as a source for omega-3 oils. This research is a well-designed fish trial and the experiments performed in this study are easy to understand and the manuscript is a well-written paper.
The pdf-file was uploaded in Ouriginal for control of plagiarism. The hole document had a similarity of 35 %, mainly from a document at DTU. I will recommend the authors to have a look at the report that is added in this review.
Some comments:
1) Be consequent of naming e.g. FAs. In the introduction the short name of a FA is written as C18:0 and at Line 242 and 245 the letter C is excluded; 18:2.
2) I will recommend that information of the number of samples replicates used is added in the figure texts.
3) Go through the manuscript and fix the use of dash and hyphens.
4) Line 38 – 40: the reference in this sentence is from 2007. We are now in 2022 so I hope that the authors can find a refence that is not that old.
5) Line 112: the unit of the length of the GC column used seen to be strange, I believe you have used a 10 m long column?
6) Line 112: I would like to know which standard mix from Sigma that was used, or include the FAMEs that are in thus mix.
7) Line 119: Use big letter of litre; L in mL and not ml.

Author Response
Thank you for the nice comments. We have addressed all the comments as detailed below:
- The pdf-file was uploaded in Ouriginal for control of plagiarism. The hole document had a similarity of 35 %, mainly from a document at DTU. I will recommend the authors to have a look at the report that is added in this review. Reply: The document at DTU is the master thesis of 2 students (co-authors of this manuscript). This manuscript is based on the thesis, so it is difficult to avoid some overlap, despite the fact that most of the text has been substantially rewritten. Particularly the discussion of the results has been rewritten.
-
Be consequent of naming e.g. FAs. In the introduction the short name of a FA is written as C18:0 and at Line 242 and 245 the letter C is excluded; 18:2. Reply: Done
-
I will recommend that information of the number of samples replicates used is added in the figure texts. Reply: A table has now been included under materials where the number of all replicates from all species is shown for the different sampling times
- Go through the manuscript and fix the use of dash and hyphens. Reply: Done
- Line 38 – 40: the reference in this sentence is from 2007. We are now in 2022 so I hope that the authors can find a refence that is not that old. Reply: Although the reference is 15 years old it is still representative for the current situation. We cannot find any newer sources of information that states the fish oil export specified as for human consumption. However, we have found data from 2019, which shows that Peru's total export of fish oil for all purposes constitutes 38 % of the global export in USD, whereas that of Chiles is worth 13 % of the total export. These numbers fit quite well with those used in the manuscript, so we have left the reference as it is.
- Line 112: the unit of the length of the GC column used seen to be strange, I believe you have used a 10 m long column? Reply: Thanks for spotting this mistake. It has been corrected
- Line 112: I would like to know which standard mix from Sigma that was used, or include the FAMEs that are in thus mix. Reply: we use the mixture Nu-Check-prep GLC 714. This information has been added in l. 114. Supplier name has been corrected.
- Line 119: Use big letter of litre; L in mL and not ml. Reply: Corrected.
Reviewer 2 Report
This manuscript (MS) was focused in the new source of oils rich in omega-3 PUFA obtained from the livers of different lean fish species. The results are very interesting, however, some points could be addressed to improve the MS.
It is important to define some terms because they are mentioned in relative terms (low; high) but there is no amount of fat is mentioned as a limit.
- what is the meaning of lean fish species (%fat), especially in line 51, where MS says: "White fish species are lean and low in fat in the flesh, but deposit a high content of oil and fat in the liver."
- What is the meaning of white fish species (% fat)
- What is the meaning of fatty fish species (% fat)
- what is the meaning of high content of oil and fat in the liver (% fat)
In this context, it would be important to include a known source of omega-3 for comparative purposes.
Introduction: In this section, it would interesting to include the estimated tons of liver to be processed and also the potential amount of oil that could be recover.
Results and discussion: This section lacks of discussion. Liver lipases effect on FFA and their activity during storage. Furthermore, what is the nutritional value of the omega-3 in those livers? in comparison of other sources of omega-3.
Conclusions: MS says: "Cod and hake and to some extent ling had EPA and DHA levels, PV and FFA content which are within the recommended levels for fish oil for human consumption" . This sentence is confusing, and lacks important data. MS should be explicit to indicate what is the recommendation (mg per day, cite) and what is the range of omega-3 in the corresponding liver or livers.
Minor comments:
- line 83. DTU meaning.
- labels in Fig. 2 are small.
Author Response
Thanks for the useful comments. We have addressed them as follows:
- what is the meaning of lean fish species (%fat), especially in line 51, where MS says: "White fish species are lean and low in fat in the flesh, but deposit a high content of oil and fat in the liver." Reply: Lean fish are generally defined as fish species with a fat content < 2 g/100 g. This has been added in l. 51.
- What is the meaning of white fish species (% fat). Reply: White fish live on or near the seafloor, in contrast to the the oily or pelagic fish which live away from the seafloor. White fish have flakier white or light-coloured flesh. As mentioned in the manuscript white fish are lean and the fat content has been added as mentioned above.
- What is the meaning of fatty fish species (% fat). Reply: Fatty fish has a fat content > 8 g/100 g. Info has been added in l. 49
- what is the meaning of high content of oil and fat in the liver (% fat) Reply: This is a general statement across all fish species, but since we did not know the exact fat content of the selected species before the study started it is not possible to give a value here.
- In this context, it would be important to include a known source of omega-3 for comparative purposes. Reply: We don't understand the question, because cod liver oil is already mentioned as a reference in the text.
- Introduction: In this section, it would interesting to include the estimated tons of liver to be processed and also the potential amount of oil that could be recover. Reply: Yes, that could be interesting, but since the purpose of the project was to determine how much oil the livers contained it is not possible to mention this before the study was carried out. With respect to the amount of livers available, these data are also not available because all the livers are currently discarded as also mentioned in the text.
7. Results and discussion: This section lacks of discussion. Liver lipases effect on FFA and their activity during storage. Furthermore, what is the nutritional value of the omega-3 in those livers? in comparison of other sources of omega-3. Reply: We assume that the reviewer refers to the discussion about the FFA, although no line numbers are stated. We have rephrased and added a bit more text on the FFA in lines. 372-375. Regarding the question about nutritional value of omega-3 fatty acids, we are not sure if we understand the question. Omega-3 fatty acids are omega-3 fatty acids no matter where they come from, so they will have the same nutritional value. However, the lipid composition of different marine oils can vary and this could lead to differences in the nutritional effects of different marine oils. F.ex. krill oils have been shown to have additional beneficial effects on top of those of omega-3 PUFA due to their content of phospholipids. However, it is beyond the scope of this study to evaluate different nutritional effects of different marine oils.
8. Conclusions: MS says: "Cod and hake and to some extent ling had EPA and DHA levels, PV and FFA content which are within the recommended levels for fish oil for human consumption" . This sentence is confusing, and lacks important data. MS should be explicit to indicate what is the recommendation (mg per day, cite) and what is the range of omega-3 in the corresponding liver or livers. Reply: These values were stated in l. 328 and 378 with references included. We have now added the values in the conclusion, but without references as conclusions usually do not include references
line 83. DTU meaning. Reply: Technical University of Denmark. Has been corrected in the text.
- labels in Fig. 2 are small. Reply: We have enlarged the figure
Reviewer 3 Report
This article is aimed on the theme and has title "Fish liver discards as a source of long chain omega-3 polyunsaturated fatty acids.
You need to check the text and insert it into the template correctly. Pay attention to text editing. For example page No. 2.
I also recommend better describing the methodology of laboratory analysis than just quoting the methodology. In this place, too, it needs to text editing.
I consider the addition of a list of samples and the definition of their collection and analysis in relation to time (date). Beacause this missing, it is a serious problem of this article.
This belongs to this section and not to the Results section (lines 137 to 144).
But even here it is very brief and confusing. The results should also be given in numerical value, for example in tables. The graphs do not accurately define the average values.
Placing graphs at the end and weak discussion and work with formulating theses and reduces the cost of this article and the work that has been done. The experiment contains a large amount of da, but these are not apparent from the graphs and their overview in the methodology is missing.
Author Response
Thanks for the comments, we have been addressed as follows:
1.You need to check the text and insert it into the template correctly. Pay attention to text editing. For example page No. 2.
Reply: The strange formatting of page 2 occurred when we tried to fill in the information regarding citation of the manuscript in the left column. We have now deleted this information and the manuscript is now following the template.
2. I also recommend better describing the methodology of laboratory analysis than just quoting the methodology. In this place, too, it needs to text editing.
Reply: Determination of dry matter, oil content, and FFA% are standard methods that have been described intensively in the literature, so we do not think it is necessary to add more information for these methods. We have elaborated on the PV method. Details were already given for the analysis of fatty acid composition so no more information has been added.
I consider the addition of a list of samples and the definition of their collection and analysis in relation to time (date). Beacause this missing, it is a serious problem of this article.
Reply: A table providing an overview of the sample has been inserted in the materials section.
This belongs to this section and not to the Results section (lines 137 to 144).
Reply: We agree that the first 3 lines could be deleted and have done so, but we think the remaining 4 lines are important and have therefore not deleted these.
But even here it is very brief and confusing. The results should also be given in numerical value, for example in tables. The graphs do not accurately define the average values.
Reply: We don't agree with this comment. We think that the graphs provide a much better overview of the data than a table would do, because it would be a very large table that would be hard for the reader to overview.
Placing graphs at the end and weak discussion and work with formulating theses and reduces the cost of this article and the work that has been done. The experiment contains a large amount of da, but these are not apparent from the graphs and their overview in the methodology is missing.
Reply: The placement of the graphs at the end of the manuscript was doen in order to follow the template from Foods. We think that the PCA and the graphs provide the best overview of all the many data. Such an overview cannot be obtained with a table. The scores plot of the first PCA shows all the many samples that the study was based on. The other reviewers have not commented that the discussion is weak and the comment from this reviewer is not very specific and therefore very difficult to do anything about.
Round 2
Reviewer 2 Report
Major comments:
I kept the previous numbers:
2.- What is the meaning of white fish species (% fat). Reply: White fish live on or near the seafloor, in contrast to the the oily or pelagic fish which live away from the seafloor. White fish have flakier white or light-coloured flesh. As mentioned in the manuscript white fish are lean and the fat content has been added as mentioned above. New comment: I suggest to include this sentence (or something similar) in the introduction. This sentence will clarify the "terminology" of the manuscript (white-lean fish) and will avoid confusing or misleading and even it would improve future citations.
4.- what is the meaning of high content of oil and fat in the liver (% fat) Reply: This is a general statement across all fish species, but since we did not know the exact fat content of the selected species before the study started it is not possible to give a value here. New comment: The problem is still here, because MS said "high", so what is "high content of oil and fat in the liver"; high is not a value. It should be clarified using data from literature.
5.- In this context, it would be important to include a known source of omega-3 for comparative purposes. Reply: We don't understand the question, because cod liver oil is already mentioned as a reference in the text. New comment: I mean an external reference using literature data for comparative purposes.
8. Conclusions: MS says: "Cod and hake and to some extent ling had EPA and DHA levels, PV and FFA content which are within the recommended levels for fish oil for human consumption" . This sentence is confusing, and lacks important data. MS should be explicit to indicate what is the recommendation (mg per day, cite) and what is the range of omega-3 in the corresponding liver or livers. Reply: These values were stated in l. 328 and 378 with references included. We have now added the values in the conclusion, but without references as conclusions usually do not include references. New comment: I don't mean this section needs references or cites, I mean that the conclusion should be improved using some sentence such as: considering FAO recommendations is xxxxx, and other values such as PV are below xxxxx, the livers from xxxx would be a interesting source of w-3. This inclusion will improve the section and even it would be more appropriate for future citations.
Minor comments
Table 1.- Some cells indicate 0, others empty. Please put 0 to homogenize.

Author Response
We have addressed the comments as follows. Numbers refer to the numbers used by the reviewer:
2. We have added a similar sentence in the introduction in line 51.
4. An example for cod liver and a reference have been added in line 54.
5. In line 446, we refer to values for commercial cod liver oils that are reported in the Frida food database. This data base is based on samples collected from Danish stores and analysed by the Danish Food Safety Authorities. We find that this is a highly relevant external reference.
8. We have restructured the conclusion a bit and added one more concluding sentence to address this comment.
Table 1 has been corrected as suggested
Reviewer 3 Report
I don't have comments about the manuscript of this article.
Author Response
There were no comments to address